# Performance of a Surrogate SARS-CoV-2-Neutralizing Antibody Assay in Natural Infection and Vaccination Samples

**DOI:** 10.3390/diagnostics11101757

**Published:** 2021-09-24

**Authors:** Kwok-Hung Chan, Ka-Yi Leung, Ricky-Ruiqi Zhang, Danlei Liu, Yujing Fan, Honglin Chen, Kwok-Yung Yuen, Ivan Fan-Ngai Hung

**Affiliations:** 1Department of Microbiology, Li Ka Shing Faculty of Medicine, University of Hong Kong, Hong Kong, China; joy2ky@hku.hk (K.-Y.L.); hlchen@hku.hk (H.C.); kyyuen@hku.hk (K.-Y.Y.); 2State Key Laboratory for Emerging Infectious Diseases, Li Ka Shing Faculty of Medicine, University of Hong Kong, Hong Kong, China; 3Carol Yu Centre for Infection, Li Ka Shing Faculty of Medicine, University of Hong Kong, Hong Kong, China; 4Department of Medicine, Li Ka Shing Faculty of Medicine, University of Hong Kong, Hong Kong, China; rickychang87@gmail.com (R.-R.Z.); liudl@hku.hk (D.L.); u3571448@connect.hku.hk (Y.F.)

**Keywords:** COVID-19, SARS-CoV-2, neutralizing antibodies, surrogate neutralization assay, live virus microneutralization

## Abstract

Severe acute respiratory syndrome coronavirus 2 (SARS-CoV-2)-neutralizing antibody (NAb) production is a crucial humoral response that can reduce re-infection or breakthrough infection. The conventional test used to measure NAb production capacity levels is the live virus-neutralizing assay. However, this test must be conducted under biosafety level-3 containment. Pseudovirus or surrogate NAb tests, such as angiotensin-converting enzyme 2 inhibition tests, can be performed under level-2 containment. The aim of this study was to evaluate the performance of a surrogate SARS-CoV-2 NAb assay (sNAb) using samples from naturally infected individuals and vaccine recipients in comparison with the live virus microneutralization assay (vMN). Three hundred and eighty serum samples which were collected from 197 patients with COVID-19, 96 vaccine recipients and 84 normal individuals were analyzed. Overall, the sensitivity, specificity, positive predictive value, and negative predictive value of the sNAb (iFlash-2019-NAb assay, Shenzhen, China) were 97.9%, 94.9%, 98.2%, and 93.8%, respectively. Agreement for the assay relative to vMN for naturally infected individuals and vaccine recipients were 98.5% and 93.9%, respectively. A correlation analysis between sNAb and the vMN for both of these groups yielded an R2 value of 0.83. The iFlash RBD NAb assay is found to be sensitive and reliable for neutralizing antibody measurement in patients with the 2019 coronavirus disease and those who have been vaccinated against it.

## 1. Introduction

Severe acute respiratory syndrome coronavirus 2 (SARS-CoV-2) has been causing a global pandemic for more than 1 year, which as of the end of July 2021 has affected about 200 million people and killed more than 4 million people [1]. SARS-CoV-2 infects cells by binding the receptor-binding domain (RBD) in the S1 subunit of its trimeric spike proteins to angiotensin-converting enzyme 2 (ACE2) on target cell surfaces [2]. Antibody binding to the RBD can prevent SARS-CoV-2 from entering the nasal epithelium, and alveolar macrophages recognize neutralized viruses and apoptotic cells and clear them by phagocytosis [3]. Such neutralization is crucial to reduce reinfection or breakthrough infection. Vaccines against SARS-CoV-2 infection were developed and made available in Hong Kong in early 2021 [4]. The Sinovac (CoronaVac) and Sinopharm (BBIBP-51 CorV) products were the first inactivated vaccines developed following the 2019 coronavirus disease (COVID-19) outbreak [5,6]. When the inactivated virus enters the human body, it induces an immune response but has no pathogenicity. The Moderna (mRNA-1273) and BNT162b2 (Comirnaty, BNT162b2-Pfizer) vaccines were developed using a platform based on mRNA encoding of the viral protein which is encapsulated in lipid nanoparticles [7]. Viral vector vaccines, such as those produced by AstraZeneca (AZD1222) and Gamaleya (Sputnik V), were developed using a platform based on vectors derived from an adenovirus and the recombined spike gene of SARS-CoV-2 [8]. Approximately 4 billion people had been vaccinated worldwide at the end of July 2021 [1]. In addition, the transfusion of COVID-19 convalescent plasma to patients with severe COVID-19 has been shown to reduce mortality [9]. This treatment strategy is based on high neutralizing antibody (NAb) titers in donor plasma from previously infected individuals.

Evaluation of the neutralizing capacity of anti-SARS-CoV-2 antibodies is important because such capacity represents real protective immunity. Live virus neutralization assays are the conventional standards for the determination of neutralizing antibody titers, but they involve live amplified virus and must be conducted under biosafety level-3 containment. Pseudovirus or ACE2 neutralizing tests are more convenient, as they can be performed under level-2 containment [10]. Thus, a reliable, sensitive, and specific surrogate neutralization assay for SARS-CoV-2 is needed. In this study, the iFlash-2019-nCoV NAb assay (Shenzhen YHLO Biotech Co. Ltd., Shenzhen, China), a surrogate SARS-CoV-2 RBD NAb assay, was evaluated. This assay had been used to measure neutralizing antibodies in COVID-19-convalescent individuals [11] and BNT162b2 vaccine recipients [12], but its performance and correlation with live virus neutralization findings had not yet been examined. Thus, its performance was evaluated using serum samples collected from naturally infected patients, vaccine recipients (COVID-19-naïve), and healthy individuals (COVID-19-naïve and COVID-19 vaccine-naïve) in this study.

## 2. Materials and Methods

### 2.1. Sample Collection and Ethical Approval

In total, 380 serum samples were collected from 197 patients with COVID-19 infection, 96 vaccine recipients and 84 normal individuals. Ninety-nine serum samples were collected from the 96 vaccine recipients (81 BNT162b2 and 15 Sinovac) who did not have SARS-CoV-2 infection before vaccination; three samples were collected before vaccination with BNT162b2 (Comirnaty; Fosun–BioNTech, Pfizer, German) or Sinovac (Coronavac, Sinovac Life Sciences, Beijing, China). The blood samples from COVID-19 patients were taken at one-month post-infection, while blood samples from vaccine recipients were taken at day 21 (BNT162b2) or day 28 (Sinovac), and day 56 for those with completion of a second dose of the vaccine. All serum samples were aliquoted and stored at −20 °C before testing. The present study was approved by the Institutional Review Board of the University of Hong Kong/Hospital Authority Hong Kong West Cluster.

### 2.2. Chemiluminescent Microparticle Immunoassay

Testing for NAbs against the SARS-CoV-2 RBD was performed using the new version of the iFlash-2019-nCoV NAb kit, a one-step competitive chemiluminescence immunoassay on the iFlash 1800 analyzer (Shenzhen YHLO Biotech Co., Ltd., Shenzhen, China) according to the manufacturer’s instructions. Briefly, the serum samples were placed on a sample rack in the sample loading area, and a reagent pack with 2019-nCoV RBD antigen (30KD)-coated paramagnetic microparticles and acridinium ester-labeled ACE2 conjugate were placed in the reagent loading area. The iFlash system performs all functions automatically. The signal from the chemiluminescent reaction is measured, and the results are determined using a calibration curve. The new version of the iFlash-2019-nCoV assay was introduced in early July 2021. The cut-off value for seropositivity is 15 AU/mL, and the maximum measurable value is 800 AU/mL. Values between >9 and <15 were considered to be in the “gray zone” or “indeterminate”. The lower and upper limits are set to <4 AU/mL and >800 AU/mL, respectively.

### 2.3. Live Virus Microneutralization

vMN was performed under biosafety level-3 containment, as described previously [13]. Briefly, it was performed in a 96-well plate. The serum samples were two-fold diluted serially, starting from 1:10 with minimum essential medium (Gibco, Green Island, NY, USA), then mixed with 100 median tissue culture infectious doses of SARS-CoV-2 and incubated at 37 °C for 1 h. Then, the mixture was added to VeroE6 cells and incubated at 37 °C and 5% CO_2_ for 72 h. The cytopathic effect was determined by examination under inversion microscopy. The vMN antibody titer was the highest dilution with 50% inhibition of the cytopathic effect, standardized to the World Health Organization’s International Standard for SARS-CoV-2 immunoglobulin (human) (National Institute for Biological Standards and Control code 20/136) [14]. vMN positivity was defined as titer ≥10 (31.25 IU/mL).

### 2.4. Statistical Analysis

The diagnostic sensitivity, specificity, positive predictive value (PPV), and negative predictive value (NPV) of the assays against the positive references and agreement between assays were calculated using VassarStats (Poughkeepsie, NY, USA, http://vassarstats.net, accessed on 4 August 2021). A simple linear regression and a Pearson correlation were computed to assess the potential association between sNAb titers and antibody titers obtained using vMN.

## 3. Results

In total, 380 serum samples were collected from 197 patients with COVID-19 (112 men, 85 women with an age range of 18–82 years), 96 vaccine recipients (46 men, 50 women with an age range of 22–71) and 84 normal individuals (44 men, 40 women with an age range of 23–50). Ninety-nine serum samples were collected from the 96 vaccine recipients (81 BNT162b2 and 15 Sinovac); three samples were collected before vaccination with BNT162b2 (*n* = 81) or Sinovac (*n* = 15). In COVID-19 patients and vaccine recipients, 288 serum samples, comprising 188 samples from patients with COVID-19 and 90 samples from vaccine recipients, were vMN positive. The patient and vaccine recipient groups each had nine vMN-negative samples. All normal samples were vMN negative. Overall, the sensitivity, specificity, PPV, and NPV of the iFlash-2019-nCoV assay were 97.9% (95% confidence interval (CI), 0.95–0.99%), 94.9% (95% CI, 0.88–0.98%), 98.2% (95% CI, 0.96–0.99%), and 93.8% (95% CI, 0.87–0.97%), respectively (Table 1). Overall concordance and non-concordance values were 97.1% and 2.9%, respectively; kappa value = 0.92 (95% CI, 0.88–0.97). The mean lower detection limit was 27.7 ± 22.9 AU/mL. When the subgroup of recovered patients was analyzed, a good correlation between the mean of surrogate Nab and vMN (*R*^2^ = 0.830) was observed (Figure 1). When the subgroup of vaccine recipients was analyzed, surrogate NAb levels yielded a similar correlation between surrogate Nab and vMN (R2 = 0.829) (Figure 2).

The mean surrogate NAb and vMN values in a subset of recipients who had received two doses of BNT162b2 (*n* = 49) or Sinovac (*n* = 15) were analyzed. The mean ± standard deviation (SD) of surrogate NAb values for samples from BNT162b2 and Sinovac recipients were 643.1 ± 244.1 AU/mL and 47.7 ± 37.1 AU/mL, respectively, whereas the mean ± SD vMN titers for these groups were 47.7 ± 37.1 and 28.9 ± 2.5, respectively (Table 2).

## 4. Discussion

ACE2 serves as the cell entry point for coronaviruses including HCoV-NL63, SARS-CoV, and SARS-CoV-2 [15], and can be used in surrogate neutralization assays. In this study, the iFlash RBD NAb assay showed good sensitivity and specificity for the measurement of SARS-CoV-2 NAbs in patients with natural infection and vaccine recipients. Its results correlated well with those of vMN in these groups. The maximum measurable value of iFlash RBD NAb assay is 800 AU/mL, which is equivalent to the mean vMN titer of 640 in this study (Figure 1 and Figure 2). This indicates that the iFlash RBD NAb assay has a narrower dynamic range than vMN for the detection of SARS-CoV-2-neutralizing antibody.

Comparison with vMN results revealed that the mean lower detection limit for the iFlash RBD NAb assay was 1.8 times higher than the limit according to manufacturer’s instruction. We suggest that the diagnostic cut-off for this assay be set at 27.7 AU/mL. The AU/mL-to-IU/mL conversion factor for the reporting of iFlash RBD NAb results is 2.4. Thus, the mean lower detection limit was 66.5 ± 55.0 IU/mL.

A recent study showed good (97.2%) concordance of old-version iFlash RBD NAb assay results with pseudovirus neutralization test (pVNT) results [12]. In another study, the old-version iFlash RBD NAb assay was used to measure NAbs in a cohort of 171 patients who had had COVID-19 at post-infection of 4–11 months, and 78.1% (164/210) of the specimens showed NAb positivity [11]. The surrogate NAb results from the present study are in agreement with findings using surrogate virus neutralization (sVNT) from a recent study [16].

The limitation of this study is that patients with other non-SARS-CoV-2 viral infection including coronaviruses are not tested. A follow-up study should be included with these patient samples to clarify its specificity in future. In the present study, we showed that the iFlash RBD NAb assay is sensitive and reliable for NAb measurement in patients with the 2019 coronavirus disease and those who have been vaccinated against it.

## Figures and Tables

**Figure 1 diagnostics-11-01757-f001:**
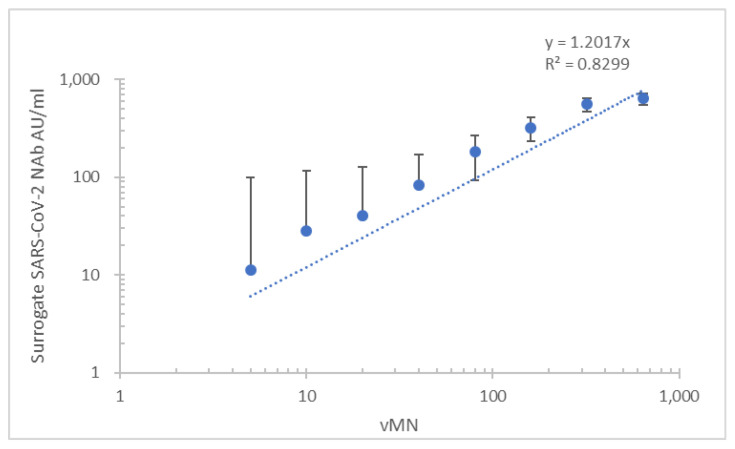
Correlation of results of the new version of the iFlash-2019-nCoV assay and vMN for the analysis of serum samples from patients with COVID-19.

**Figure 2 diagnostics-11-01757-f002:**
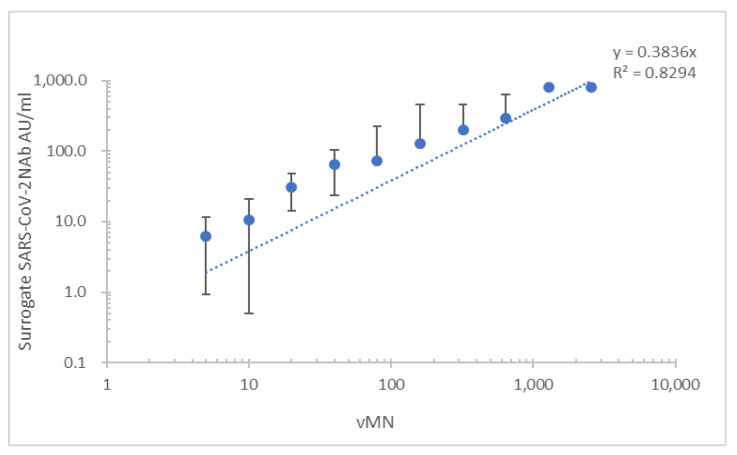
Correlation of results of the new version of the iFlash-2019-nCoV assay and vMN for the analysis of serum samples from recipients of the BNT162b2 or Sinovac vaccine.

**Table 1 diagnostics-11-01757-t001:** Performance of the iFlash-2019-nCoV assay.

	Sensitivity	Specificity	Positive Predictive Value	Negative Predictive Value
Surrogate NAb	97.9%	94.9%	98.2%	93.8%
(95% CI, Lower-Upper limit)	(0.95–0.99)	(0.88–0.98)	(0.96–0.99)	(0.87–0.97)

**Table 2 diagnostics-11-01757-t002:** Antibody response at day 56 after completion of two vaccine doses.

Vaccine	Number	Mean of sNAb (AU/mL)	SD	GM of vMN (Titers)	SD
BNT162b2	49	643.1	244.1	248.1	3.2
Sinovac	15	47.7	37.1	28.9	2.5

## Data Availability

The data used to support the findings of this study are included within the article.

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
