# Peer review of "Performance of a Surrogate SARS-CoV-2-Neutralizing Antibody Assay in Natural Infection and Vaccination Samples"

_diagnostics, 2021, doi:10.3390/diagnostics11101757_

Round 1

Reviewer 1 Report

Authors tested sera from COVID-19 convalescent and from vaccinated persons.

Sera from control persons were also unsed.Sera were used to detect antibodies againt the RBD of SARS-CoV-2 using two different tests a NA-immunoassay and a live virus microneutralisation assay.Authors found that the tests gave comparable result but the amount of antibodies induced by the BNT162b2-Pfizer vaccine was higher than that induced by the Sinovac vaccine.The results suggest that the surrogate test to determine neutralizing antibodies can be used replacing the vMN.

Comments:

1.the definition of "neutralizing" antibody can be misleading.Infact it suggests that the antibodies measured in the sera of patients and of vaccinees have the capacity to inhibit viral attachment not only to the vero cells but also to the nasal epithelium also in vivo.

Furthermore viral neutralisation may mean that the antibody binds to a virus in the blood circulation and by this mean it helps to clear the virus by the tissue macrophages.This should be stated in the introduction.

It also should made clear that the vaccines contain different delivery tools (e.g. lipid nanoparticles) which may have a booster effect even without the Spike-RNA.Infact neutralizing antibodies have been found in the serum of two  controls.

2.it is therefore very important to have RBDs from other corona viruses (this is mentioned as limitation at the end of the discussion) as negative controls.

3.auhtors avoides to report about the mean age and the range of age of the controls (Table 1).

4.authors also do report the time at which bloood war taken from vaccinees and from convalescent patients.

5. ashort description of the antigen used for the immunoassay would be important for the readers

Author Response

Thank you for your valuable comments:

1.the definition of "neutralizing" antibody can be misleading. In fact it suggests that the antibodies measured in the sera of patients and of vaccinees have the capacity to inhibit viral attachment not only to the vero cells but also to the nasal epithelium also in vivo.

Furthermore viral neutralisation may mean that the antibody binds to a virus in the blood circulation and by this mean it helps to clear the virus by the tissue macrophages.This should be stated in the introduction.

It also should made clear that the vaccines contain different delivery tools (e.g. lipid nanoparticles) which may have a booster effect even without the Spike-RNA. Infact neutralizing antibodies have been found in the serum of two controls.

Author response: We agree with the reviewer’s comments and rewrite the sentences in lines 39-42 “Antibody binding to the RBD can prevent SARS-CoV-2 from entering to nasal epithelium and also bind to the virus in the blood circulation which helps to clear the virus through the tissue macrophages.” and in line 46, “When the inactivated virus enters the human body, it induces an immune response but has no pathogenicity” and line 49 “which is encapsulated in lipid nanoparticles” has been added in the sentences.

2.it is therefore very important to have RBDs from other corona viruses (this is mentioned as limitation at the end of the discussion) as negative controls.

Author response:  In lines 169-170, we add “A follow-up study should be included with these patient samples to clarify its specificity in future”.

3.auhtors avoides to report about the mean age and the range of age of the controls (Table 1).

Authors response: Table 1 was deleted as requested by Reviewer 2. In line 119, we add “84 normal individuals (43 men, 41 women; with an age range of 22-53)”.

4.authors also do report the time at which bloood war taken from vaccinees and from convalescent patients.

Authors response: We add “The blood samples from COVID-19 patients were taken at one-month post-infection; while blood samples from vaccine recipients were taken at day 21 (BNT162b2) or day 28 (Sinovac), and day 56 for those completion of second dose of vaccines.” in line 77-81”.

5.a short description of the antigen used for the immunoassay would be important for the readers

Authors response: This is a commercial immunoassay. The manufacturer did not disclose much information about SARS-CoV-2 RBD protein being used in the assay kit. We have added RBD antigen (30KD) in line 89. The SARS-CoV-2 RBD has also been described in line 37, “the receptor-binding domain (RBD) in the S1 subunit of its trimeric spike proteins”.

All changes have been highlighted in revised manuscript.

Reviewer 2 Report

The article Performance of a Surrogate SARS-CoV-2 Neutralizing Antibody Assay in Natural Infection and Vaccination Samples is quite well written and structured. A few issues might be addressed before acceptance.

The most important is the evaluation of the titer between two vaccine that is out of the purpose of the study.  A contradiction in the discussion might be addressed  and explained. 

L29-31 Out of the purpose of the study, I advice to remove these sentence.

L40-42 SARS Cov 2 infect cells by binding the RBD to ACE2 on infected cell surface (if this is the way virus infects cells, they cannot by already infected)

L60 sophisticated is not necessary.

L68 had not yet been examined

L111-112 Seems out of the purpose of the study. 

L75-77 L118 How many before each of the vaccine?

L121 Describe and discuss the outcome of these two samples

137 Table 1 are not necessary as it is, the first two columns are sufficient. You do not discuss gender or age. Plus mean is not good to describe age distribution, the median is better. 

L143-145 Out of purpose. 

L161-163 In contradiction with the conclusion

L168-170 Out of purpose. 

Author Response

Thank you for your valuable comments:

L29-31 Out of the purpose of the study, I advised to remove these sentence.

Author response: As advised, the sentences have been removed.

L40-42 SARS Cov 2 infect cells by binding the RBD to ACE2 on infected cell surface (if this is the way virus infects cells, they cannot by already infected)

Author response: As advised, the sentence has been re-written as” SARS-CoV-2 infects cells by binding the receptor-binding domain (RBD) in the S1 subunit of its trimeric spike proteins to angiotensin-converting enzyme 2 (ACE2) on target cell surfaces”(lines 37-39).

L60 sophisticated is not necessary.

Author response: As advised, the word has been removed.

L68 had not yet been examined

Author response: As advised, the sentence has been corrected.

L111-112 Seems out of the purpose of the study.

Author response: As advised, the sentence has been removed.

L75-77 L118 How many before each of the vaccine?

Author response: The number of BNT162b2 and Sinovac vaccine recipients are 81 and 15 respectively. The numbers have been added in the sentences lines 74 and 122.

L121 Describe and discuss the outcome of these two samples

Author response: We have made a mistake that all normal samples should be vMN negative.  In fact, these two samples are surrogate NAb positive. We have corrected the sentence “All normal samples were vMN negative (line 125).

L143-145 Out of purpose.

Author response: As advised, the sentence has been removed

L161-163 In contradiction with the conclusion

Author response: The difference which may be due to the narrow dynamic range of Surrogate NAb assay is within acceptable SD range. This is why we suggest vMN should be used for confirmation if necessary. The sentence has been removed to avoid misunderstanding.

All changes have been highlighted in revised manuscript.

Round 2

Reviewer 1 Report

the second part of the sentence introduced at the befinning of the introduction is only true for viruses which reach the general circulation.This has not been demonstrated for SARS-Co-2 jet.Therefore this should be clarified.Infact it should be made clear that this is only an hypothesis

Author Response

Thank you for your valuable comments.

Authors response: We agree the comments and rewrite the sentence (line 39) ”Antibody binding to the RBD can prevent SARS-CoV-2 from entering to nasal epithelium; and alveolar macrophages recognize neutralized viruses and apoptotic cells and clear them by phagocytosis”. A reference was cited